# The Influence of CLBP Window Size on Urban Vegetation Type Classification Using High Spatial Resolution Satellite Images

**Zhou Chen [1], Xianyun Fei [1,\*], Xiangwei Gao [1], Xiaoxue Wang [1], Huimin Zhao [1], Kapo Wong [2], Jin Yeu Tsou [3,4] and Yuanzhi Zhang [4,5]**

[1]  School of Geomatics and Marine Information, Jiangsu Ocean University, Lianyungang 222002, China; 2017220124@jou.edu.cn (Z.C.); 2007000057@jou.edu.cn (X.G.); 2017210016@jou.edu.cn (X.W.); 2018224066@jou.edu.cn (H.Z.)

[2]  Department of Systems Engineering and Engineering Management, City University of Hong Kong, Kowloon, Hong Kong, China; kpwong42-c@my.cityu.edu.hk

[3]  Department of Architecture and Civil Engineering, City University of Hong Kong, Hong Kong, China; jinyeutsou@cuhk.edu.hk

[4]  Faculty of Social Science and Asia-Pacific Studies Institute, The Chinese University of Hong Kong, New Territories, Hong Kong, China; yuanzhizhang@cuhk.edu.hk

[5]  School of Marine Science, Nanjing University of Information Science and Technology, Nanjing 210044, China

\*  Correspondence: 2007000058@jou.edu.cn

**Abstract:** Urban vegetation can regulate ecological balance, reduce the influence of urban heat islands, and improve human beings' mental state. Accordingly, classification of urban vegetation types plays a significant role in urban vegetation research. This paper presents various window sizes of completed local binary pattern (CLBP) texture features classifying urban vegetation based on high spatial-resolution WorldView-2 images in areas of Shanghai (China) and Lianyungang (Jiangsu province, China). To demonstrate the stability and universality of different CLBP window textures, two study areas were selected. Using spectral information alone and spectral information combined with texture information, imagery is classified using random forest (RF) method based on vegetation type, showing that use of spectral information with CLBP window textures can achieve 7.28% greater accuracy than use of only spectral information for urban vegetation type classification, with accuracy greater for single vegetation types than for mixed ones. Optimal window sizes of CLBP textures for grass, shrub, arbor, shrub-grass, arbor-grass, and arbor-shrub-grass are $3 \times 3$, $3 \times 3$, $11 \times 11$, $9 \times 9$, $9 \times 9$, $7 \times 7$ for urban vegetation type classification. Furthermore, optimal CLBP window size is determined by the roughness of vegetation texture.

**Keywords:** urban vegetation type; completed local binary pattern (CLBP); window size texture; the optimal window size; roughness

## 1. Introduction

Vegetation plays an important role in urban ecology, environment, and daily life. Urban vegetation maintains urban ecological balance, reduces the effects of urban heat islands, and improves the quality of the living environment [1–8]. In addition, it can produce social benefits [9], such as by reducing crime rates [10], improving social relationships [11,12], and boosting residential property values [13]. Accordingly, the study of urban vegetation is of great significance.

High-resolution remote sensing images are particularly suitable for urban vegetation type classification. Yu et al. [14] noted better classification accuracy with Digital Airborne Imagery System

(DAIS) images than medium-resolution images during detailed vegetation classification in Northern California. Zhang and Feng [15] reported an accuracy of 87.71% in classifying five types of urban vegetation types in Nanjing (Jiangsu province, China) using IKONOS images. Compared with medium-resolution satellite images, high-resolution images have more detailed spatial information, which is instrumental in urban vegetation type classification.

Texture is the regular variation of pixel values in a digital image space. For vegetation images, textures are actually created by changes in the pattern, species, and density of vegetation, characteristics that are all closely related to vegetation types [16,17]. As a particularly important form of spatial information, textures have been extensively used to improve the accuracy of urban vegetation type classification based on high spatial-resolution remote sensing images [14,18]. The grey level co-occurrence matrix (GLCM) texture is the most widely used in vegetation classification. Yan et al. [19] used IKONOS images to extract urban grass information in Nanjing, Jiangsu Province, China, comparing the extraction accuracy of GLCM-Contract, GLCM-Entropy, GLCM-Correlation, and GLCM-Angular-Second-Moment, and found the highest extraction accuracy, reaching 90.56%, for GLCM-Contract. Pu and Landry [20] used WorldView-2 and IKONOS images to classify urban forest tree species in Tampa, Florida, with six kinds of GLCM in texture features used through LDA and CART classification methods. Urban forest trees were divided into seven types, with the classification accuracy of WorldView-2 (2 m) exceeding that of IKONOS images (4 m). The average highest classification accuracy of WorldView-2 was 67.22%; that of IKONOS was 53.67%.

Recently it has been reported that local binary pattern (LBP) is more efficient than GLCM in remote sensing classification [21,22]. LBP, which calculates the gray value difference between a center pixel and its neighborhoods, has been used in remote sensing image classification [23–25]. Song et al. [21] compared LBP and GLCM of four land-cover types classification using IKONOS images and found that overall accuracy for LBP was 4.42% higher than for GLCM. Similarly, Vigneshl and Thyagharajan [22] reported better overall accuracy for LBP than traditional GLCM. The completed local binary patterns (CLBP) algorithm, an enhanced version of LBP [26], could fully describe the texture information of remote sensing images, unlike LBP. Li et al. [27] investigated land-use classification using CLBP textures, achieving an overall accuracy of 93.3%. Wang et al. [28] employed CLBP textures to classify coastal wetland vegetation and achieve an overall accuracy of 85.38%. However, because CLBP textures have not been widely used to classify urban vegetation types, CLBP textures' potential for urban vegetation type classification needs further exploration.

Window size is an important parameter for CLBP textures and is closely related to the accuracy of image segmentation and classification [29,30]. Numerous studies have shown that no single window size can describe the landscape characteristics of all remotely sensed objects [31–35]. To achieve higher classification accuracy, texture features of different objects should be analyzed using different window sizes. For example, GLCM textures found that window size is an important parameter in vegetation classification [36–38]. Yan et al. [19] used $3 \times 3$ and $5 \times 5$ window size GLCM textures to improve the grass extraction accuracy in Nanjing (Jiangsu province, China) and found that the $3 \times 3$ window size produced better classification results. Fu and Lin [39] extracted the loquat using $3 \times 3$, $5 \times 5$, $7 \times 7$, $9 \times 9$, and $11 \times 11$ GLCM window textures and found that the $7 \times 7$ window size produced the highest classification accuracy, 86.67%. They concluded that different window sizes have different effects on classification of different vegetation types.

Compared with GLCM, the optimal window size of CLBP textures has rarely been discussed for classification of urban vegetation types using high spatial-resolution remote sensing images. This study explores the impact of window sizes of CLBP texture extraction on classification of urban vegetation types.

The main experimental process of this paper is shown in Figure 1. The experimental process is mainly divided into remote sensing image preprocessing, CLBP window code writing, image segmentation, classification, and accuracy analysis.

In this paper, it mainly explains the function of urban vegetation in the city. The use of high-resolution remote sensing images is more suitable for the classification of urban vegetation types. In the study of urban vegetation classification, texture features play an important role in improving the classification effect. As a texture feature, CLBP has great potential in the classification of urban vegetation types, and it is of great research value to explore the influence of CLBP textures in different window sizes on the classification of urban vegetation types.

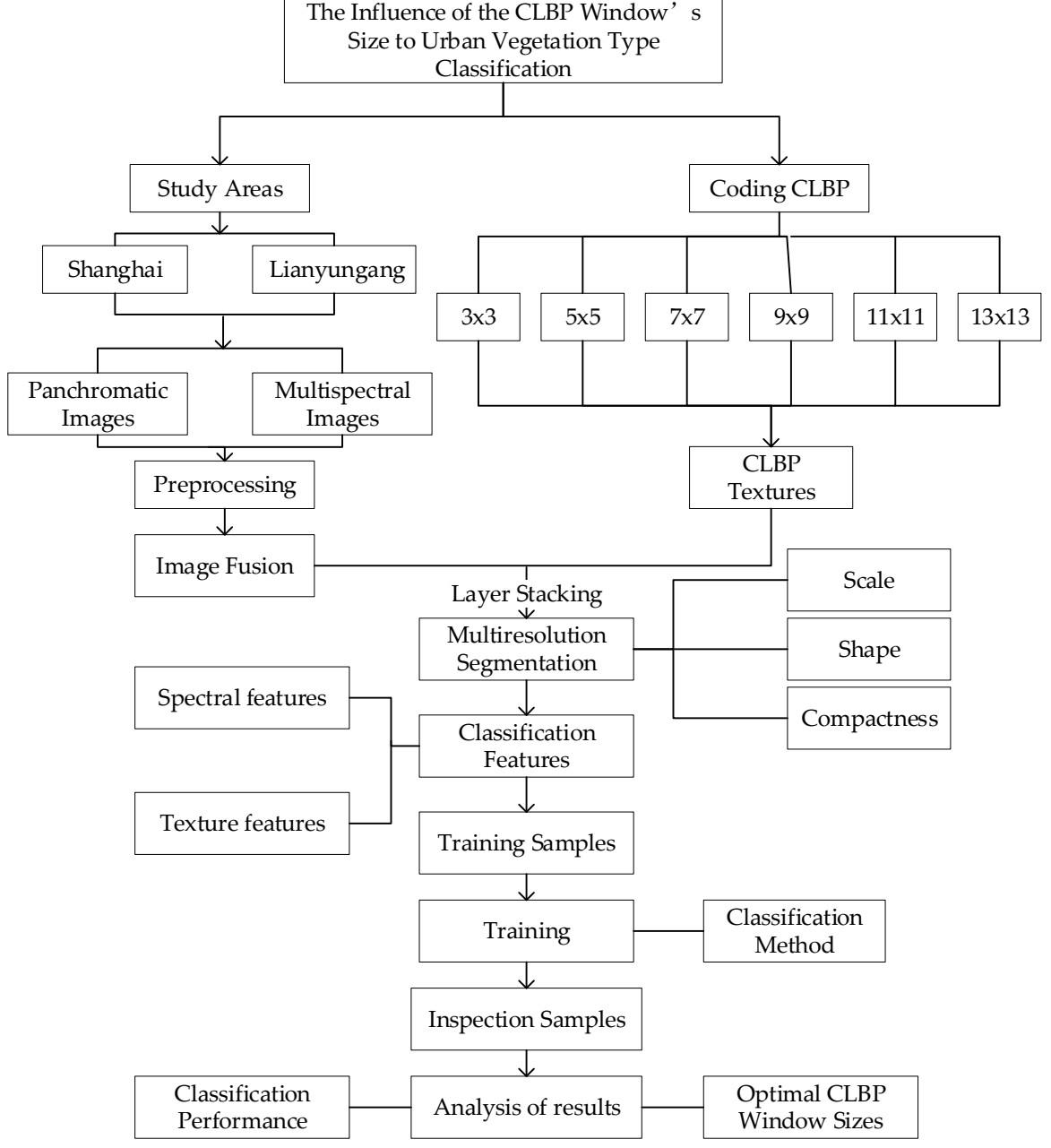

**Figure 1.** The experimental process.

## 2. Study Area and Data

### 2.1. Study Area

Shanghai, one of China's commercial capitals, features a subtropical maritime monsoon climate, abundant rainfall, advanced urban planning, and intelligent technology development, which have

led to flourishing urban vegetation and a reasonable scale layout. Urban vegetation construction in Shanghai is at the forefront in China.

Lianyungang, a coastal city in eastern China, in the northeast of Jiangsu Province, in the Huaihe River basin, has the climate characteristics of China's south and north. Its economic level and population are representative of China as a whole.

The study area of Shanghai was selected at Shanghai Jiao Tong University (Minhang Campus), with an area of 8.8 km², at 121°25′30″–121°27′35″E and 31°0′55″–31°2′50″N (see Figure 2). The vegetation type is mainly mixed with evergreen broad-leaved forest and artificial shrubs and grass. Vegetation distribution and planning are reasonable, and the growth of vegetation is prosperous. The research area of Lianyungang is nearby Haizhou Municipal Government, covering an area of 9.38 km², at 119°7′55″–119°9′55″E and 34°34′5″–34°35′35″N (see Figure 2). The vegetation types in this area are similar to Shanghai, whilst the growth of vegetation and the vegetation planning and management is worse than Shanghai. The urban vegetation planning and growth in Lianyungang is representative in China.

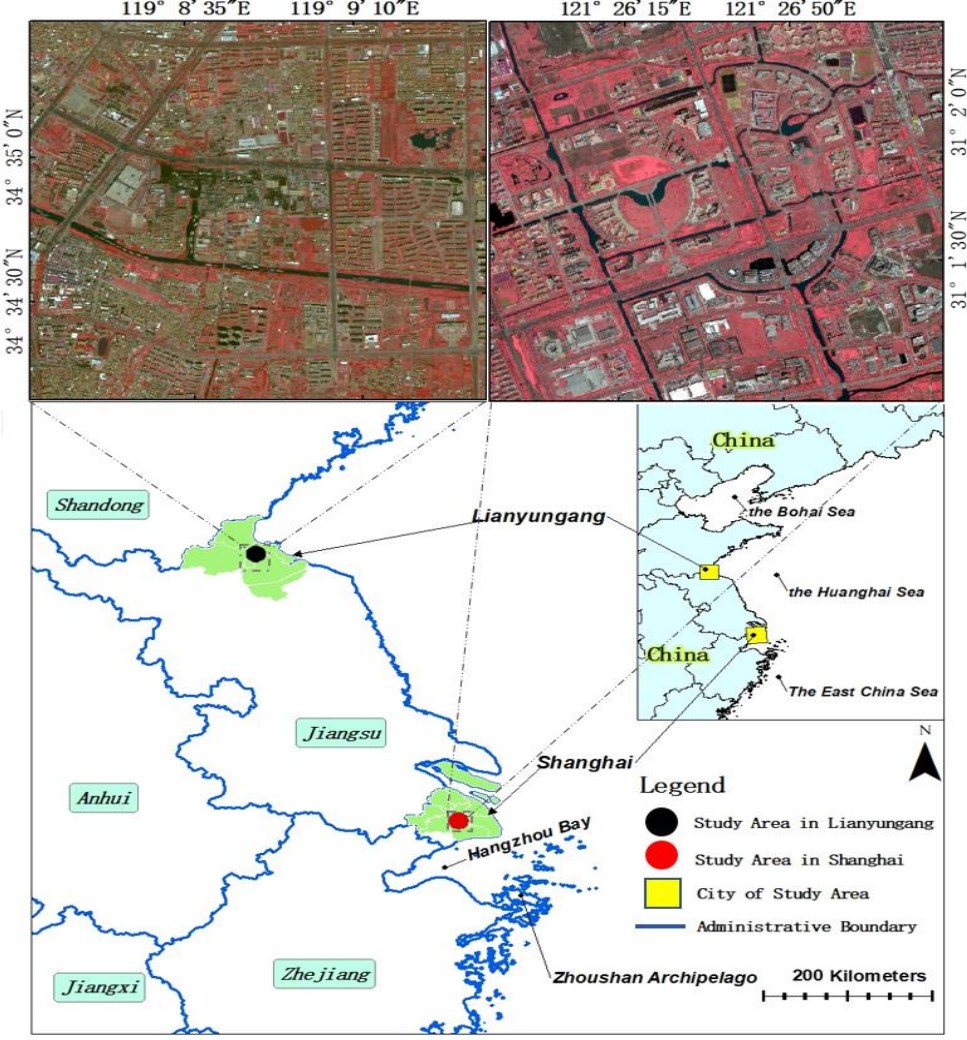

**Figure 2.** The location of the study area in Shanghai and Lianyungang, China.

*2.2. Data*

Both study areas used images from WorldView-2, acquired from April to May in 2015, without clouds or deformation; the terrain of the study areas is flat. Accordingly, the images are suitable for remote sensing vegetation classification. The images of the study areas in Shanghai and Lianyungang are

composed of multispectral bands at 2 m resolution and panchromatic bands at 0.5 m resolution. Table 1 gives relevant information about the images (https://dg-cms-uploads-production.s3.amazonaws.com/uploads/document/file/98/WorldView2-DS-WV2-rev2.pdf) and Table 2 shows the projection coordinates and geographic coordinate system of the images.

**Table 1.** WordView-2 band descriptions.

| Spectral Band | Wavelength (nm) | Spatial Resolution (m) |
| --- | --- | --- |
| Red | 630–690 | 0.46 |
| Green | 510–580 | 0.46 |
| Blue | 450–510 | 0.46 |
| NIR | 770–895 | 0.46 |

**Table 2.** The projection coordinates and geographical coordinates of the study areas.

| Coordinates | Shanghai | Lianyungang |
| --- | --- | --- |
| Projected coordinate | WGS_1984_UTM_Zone_51N | WGS_1984_UTM_Zone_50N |
| Geographic coordinate | GCS_WGS_1984 | |

In the data preprocessing process, geometric correction and atmospheric correction were first conducted for each study area. Subsequently, the GS method was used to fuse four multispectral bands and one panchromatic band. The resulting image had 0.5 m spatial resolution and multispectral information. CLBP texture features were extracted by sharpening images, with urban vegetation in the two study areas classified by spectral information and textures.

### 2.3. Image Segmentation

Image segmentation models spatial relationships and dependencies by dividing a whole image into continuous and spatially independent objects [40]. The essence of segmentation is the clustering of pixels. In continuous iterative steps, similar pixels are merged into small objects that are themselves merged into larger objects [41], solving the salt-and-pepper problem [13]. Image segmentation makes full use of images' spatial information [17] and has a high classification efficiency [40]. In this study, image objects are used to classify urban vegetation based on image segmentation.

The image was segmented using a multiresolution segmentation algorithm, a bottom-up region-merging technique [39]. Such approaches have three main parameters: scale, shape, and compactness [14]. The scale parameter determines the maximum difference of the image object by adjusting the scale parameter to determine the size of the segmentation object. The shape parameter determines the degree of difference in the shape of the segmentation object, and the compactness parameter determines the degree of fragmentation of the segmented object.

In this study, image segmentation had two steps: first segmentation of the entire image, then segmentation of the vegetation information. Over many attempts, the shape parameter was set to 0.2 and the compactness parameter to 0.6 in the first step of segmentation. After determination of shape and compactness parameters, the ESP (estimation of scale parameter) tool was used to help obtain the optimal segmentation scale, building on the idea of local variance of object heterogeneity within a scene. Peaks in the rate of change (ROC) graph indicated that imagery was segmented at the most appropriate scale [42].

The most suitable ESP scale for distinguishing vegetation in detail was 52 (Figure 3). Based on ESP, an artificial experiment (Table 3) was conducted to identify the optimal scale parameter for the first segmentation as 50.

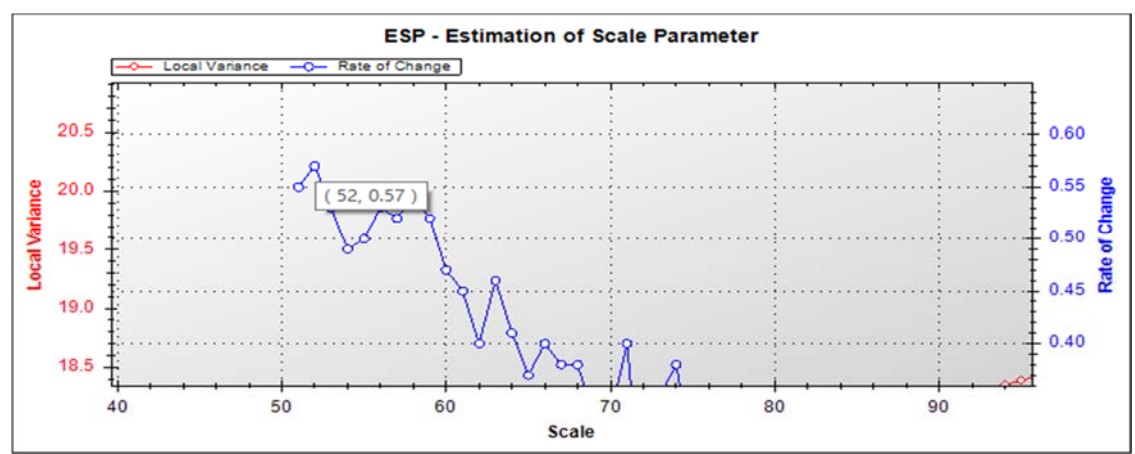

**Figure 3.** Scale parameters obtained by ESP (estimation of scale parameter).

**Table 3.** Comparison of segmentation scale effects (based on ESP).

| | Target 1 | Target 2 | Target 3 | Shape/Compactness |
|---|---|---|---|---|
| **Scale: 40** | | | | 0.2/0.6 |
| **Scale: 50** | | | | |
| **Scale: 60** | | | | |

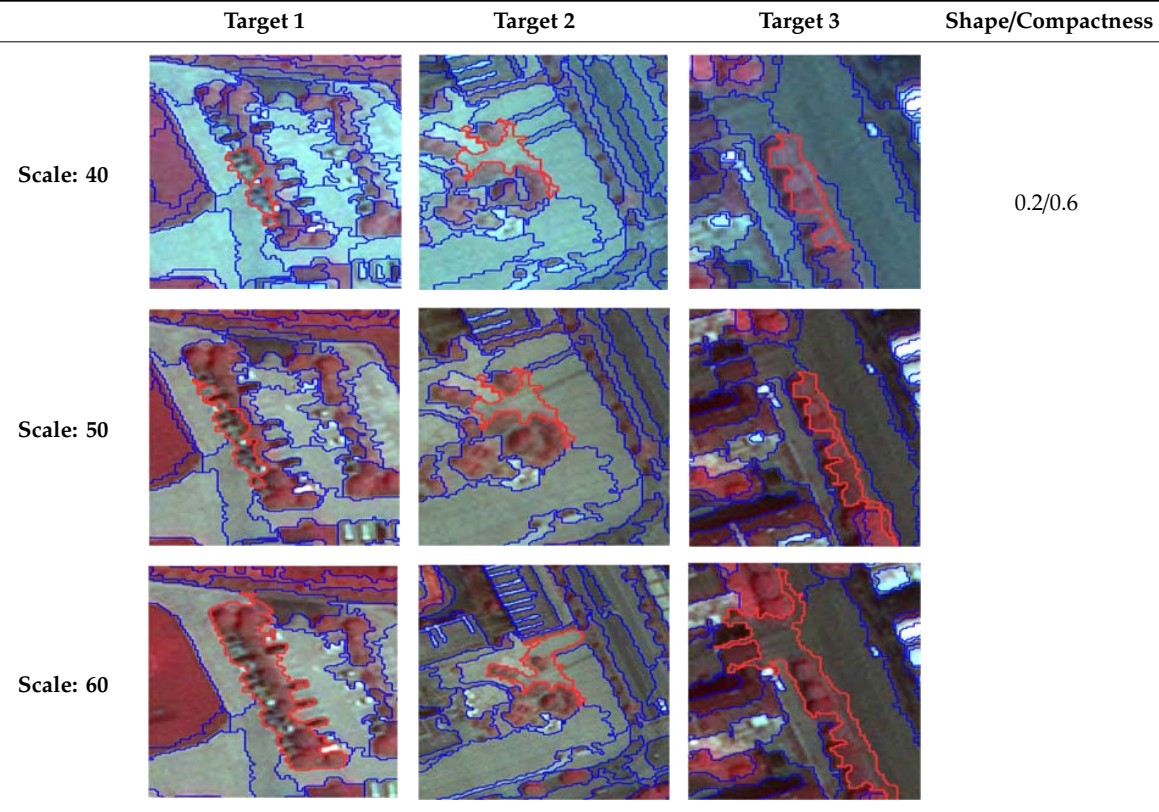

The second segmentation is based on the first step. The optimal scale parameter in the second step was 80, while shape and compactness parameters were 0.2 and 0.4. In the first segmentation, vegetation information was mainly separated from other ground objects and thus the segmentation scale became relatively small to obtain more detailed vegetation information. In the second segmentation, if the segmentation scale was too small, the vegetation would be prone to over segmentation. Therefore, the optimal scale of first segmentation was 50, and the second was 80 through the ESP and manual testing.

### 2.4. Feature Extraction

For this study, 19 inconsistent features—seven spectral features and 12 CLBP texture features—were used in each vegetation classification experiment. Spectral features included the spectral value of

four bands (red, blue, green, and NIR), max.diff, and NDVI. Texture features included the CLBP_M, CLBP_S, and CLBP_C texture features of each band (red, blue, green, and NIR) with six different window sizes. Table 4 gives detailed description of the features.

**Table 4.** Each window's features.

| Object Features | Feature Parameters | Explains |
|---|---|---|
| | Spectral values | Red, Blue, Green, NIR |
| | Brightness | Average value of Image Spectrum (as shown in Formula (1)) |
| Spectral features | NDVI | (NIR-Red)/(NIR+ Red) |
| | Max.diff | The absolute value of the difference between the maximum band gray values divided by the average brightness values (as shown in Formula (2)) |
| | CLBP_S | The signed component of Red, Blue, Green, and NIR bands (as shown in Formula (4)) |
| Texture features | CLBP_M | The size component of Red, Blue, Green, and NIR bands (as shown in Formula (5)) |
| | CLBP_C | The central pixel value binary component of Red, Blue, Green, and NIR bands (as shown in Formula (6)) |

In Formula (1), the $w_k^B$ is the brightness weight of layer k, $K$ is the number of layers, $\bar{c}_k(v)$ is the mean intensity of layer k of image object $v$ (Trimble.eCognition® Developer9.0.1 User Guide, https://geospatial.trimble.com/ecognition-trial).

$$Brightness = \frac{1}{w^B}\sum_{k=1}^{K} w_k^B \bar{c}_k(v) \qquad w^B = \sum_{k=1}^{K} w_k^B \tag{1}$$

In Formula (2), $i$, $j$ are image layers, $\bar{c}(v)$ is the brightness of image object $v$, $\bar{c}_i(v)$ is the mean intensity of layer i of image object $v$, $\bar{c}_j(v)$ is the mean intensity of layer $j$ of image object $v$. $K_B$ are layers of positive brightness weight with $K_B$, $w_k$ is the layer weight (Trimble.eCognition® Developer9.0.1 User Guide, https://geospatial.trimble.com/ecognition-trial).

$$Dax.diff = \frac{max_{i,j\varepsilon K_B}\left|\bar{c}_i(v) - \bar{c}_j(v)\right|}{\bar{c}(v)} \qquad K_B = \{k \in K : w_k = 1\} \tag{2}$$

CLBP textures were used as an improvement on and optimization of LBP—a nonparametric local texture description algorithm that incorporates a simple principle, low computational complexity, and invariant illumination [25].

$$LBP_{P,R} = \sum_{p=0}^{P-1} s\left(g_p - g_c\right)2^p, s(x) = \begin{cases} 1, & x \geq 0 \\ 0, & x < 0 \end{cases} \tag{3}$$

In Formula (3), $g_p$ is the gray value of the central pixel in the window, $g_c$ is the gray value of neighbors, $p$ is the number of neighbor pixels in the window, $R$ is the size of the window [26].

As the original LBP does not make full use of image information, Guo et al. proposed CLBP [43], which includes mainly a signed component (CLBP_S), a magnitude component (CLBP_M), and a central pixel value binary component (CLBP_C). A neighborhood's pixel value is compared with the central pixel value to obtain the corresponding symbol value of CLBP_S and magnitude value of CLBP_M. CLBP_C is obtained by comparing the central pixel value with the global average pixel value. On the basis of CLBP (CLBP_S, CLBP_M and CLBP_C), window sizes are changed to study the classification of urban vegetation types.

Based on the coding mode of LBP, different window sizes of CLBP textures were coded according to LBP. The value of 1 in CLBP_S was encoded as 1, and the value of −1 was encoded as 0. CLBP_S was the original LBP as shown in Formula (1), the value of CLBP_M was encoded by 1/0 as shown in Formula (2), and CLBP_C was a 1/0 binary grayscale image.

$$CLBP\_S = \sum_{i=0}^{i-1} S(P_s - P_c)2^i, S(P_s - P_c) = \begin{cases} 1 \to 1, x \geq 0 \\ -1 \to 0, x < 0 \end{cases} \tag{4}$$

Formula (4) shows that the encoding process of CLBP_S, in which $P_s$ is the neighborhood's pixel value of the window, $P_c$ is the pixel value of the central pixel of the window, and $S(P_s - P_c)$ is a symbol value that makes the difference between the neighborhood pixel value of the window and the central pixel value [26]. The value of 1 is positive and −1 is negative. In the encoding, the positive value is coded to 1, the negative value is coded to 0, and then the value of CLBP_S is obtained.

$$CLBP\_M = \sum_{i=0}^{i-1} M(U_s, c)2^i, M(U_s, c) = \begin{cases} 1, x \geq c \\ 0, x < c \end{cases}$$
$$U_s = |P_s - P_c| \, c = \frac{\sum_{s=0}^{s-1} U_s}{s} \tag{5}$$

Formula (5) shows that $c$ is the mean of $U_s$ for the whole image [26]. $U_s$ is the absolute value of the difference between the pixel value of the neighborhood's pixel and the central pixel value of the window. s is the number of the center pixel. The value of CLBP_M can be obtained by encoding 1/0 and converting it to decimal value.

$$CLBP\_C = C(P_s, m(c)), C(P_s, m(c)) = \begin{cases} 1, P_s \geq m(c) \\ 0, P_s < m(c) \end{cases}$$
$$m(c) = \frac{\sum_{i=0}^{i-1} P_i}{c} \tag{6}$$

Formula (6) shows that $m(c)$ is the mean of the pixel value of the whole image [26]. When $P_s \geq m(c)$, the value of CLBP_C is 1. When $P_s < m(c)$, the value of CLBP_C is 0. With the increase of CLBP window size, the locality of CLBP information and data will be lost gradually, and the theoretical window size is extended to the entire image.

*2.5. Vegetation Type Determination and Sample Selection*

The vegetation types of the study areas were divided into three single types and three mixed types: grass, shrub, arbor, shrub-grass, arbor-grass, and arbor-shrub-grass. Shrub-grass, arbor-grass, arbor-shrub-grass are abbreviated as SG, AG, and ASG, respectively. These types were decided by urban vegetation characteristics and the results of multiresolution segmentation.

Table 5 shows the vegetation types, the principles for the selection, and field photographs of vegetation types samples. Table 6 shows the typical examples under different features (spectrum, CLBP_M, CLBP_S, CLBP_C) of each vegetation type.

**Table 5.** The selection principle of vegetation type samples.

| Vegetation Types | Selection Principles | Field Photographs |
|---|---|---|
| **Grass** | The color transition is uniform and regular on the pseudo-color map, and the CLBP window texture is composed of regular, continuous, and smooth small patches. |  |
| **Shrub** | The color transition is uneven and discontinuous on the pseudo-color map, and CLBP window texture is composed of irregular, discontinuous patches of different sizes. |  |
| **Arbor** | There are obvious crowns on the pseudo-color map, and CLBP window texture presents regular, discontinuous large patches. |  |
| **SG** | It is composed of shrub and grass, and the color shows the characteristics of shrub and grass on the pseudo-color map. The CLBP window texture is composed of irregular small, medium plaques and smooth patches. |  |
| **AG** | On the pseudo-color map, there are characteristics of arbor and grass. The CLBP window texture is composed of large patches of arbor's crown and small patches or smooth parts of grass. |  |

**Table 5.** *Cont.*

| Vegetation Types | Selection Principles | Field Photographs |
|---|---|---|
| ASG | It consists of the color characteristics of arbor, shrub, and grass on the pseudo-color map. The CLBP window texture also has the texture characteristics of all vegetation types. |  |

**Table 6.** The typical examples under different features of each vegetation type.

| Vegetation Types | Spectrum | CLBP_M | CLBP_S | CLBP_C |
|---|---|---|---|---|
| Grass |  |  |  |  |
| Shrub |  |  |  |  |
| Arbor |  |  |  |  |
| AG |  |  |  |  |
| SG |  |  |  |  |
| ASG |  |  |  |  |

This paper describes two parts of vegetation samples: training samples and inspection samples. In total, 6350 vegetation samples were used: 3773 samples from Shanghai and 2577 from Lianyungang, comprising 2688 training samples and 3673 inspection samples. Table 7 shows sample selection.

**Table 7.** The sample statistics.

|  |  | Grass | Shrub | Arbor | SG | AG | ASG | Sum |
|---|---|---|---|---|---|---|---|---|
| 3 × 3 | Training | 134 | 120 | 99 | 38 | 65 | 43 | 499 |
|  | Inspection | 146 | 146 | 108 | 47 | 68 | 44 | 559 |
| 5 × 5 | Training | 93 | 88 | 79 | 53 | 55 | 47 | 415 |
|  | Inspection | 101 | 119 | 94 | 65 | 65 | 51 | 495 |
| 7 × 7 | Training | 66 | 84 | 83 | 44 | 59 | 48 | 384 |
|  | Inspection | 83 | 85 | 97 | 60 | 79 | 68 | 472 |
| 9 × 9 | Training | 66 | 75 | 77 | 65 | 57 | 38 | 378 |
|  | Inspection | 132 | 106 | 100 | 83 | 92 | 70 | 583 |
| 11 × 11 | Training | 70 | 72 | 81 | 44 | 52 | 43 | 362 |
|  | Inspection | 108 | 93 | 127 | 66 | 67 | 67 | 528 |
| 13 × 13 | Training | 47 | 49 | 50 | 41 | 38 | 47 | 272 |
|  | Inspection | 88 | 83 | 82 | 62 | 60 | 64 | 439 |
| Only Spectrum | Training | 64 | 84 | 96 | 46 | 46 | 42 | 378 |
|  | Inspection | 100 | 124 | 112 | 90 | 93 | 67 | 586 |
| Sum |  | 1298 | 1328 | 1285 | 804 | 896 | 739 | 6350 |

The experiment in this paper is unable to achieve a unified training sample size, because CLBP textures with different window sizes are added to the multiscale segmentation process. Different window sizes of CLBP textures lead to inconsistent segmentation results under the same segmentation scale, and thus this makes vegetation types more clearly segmented and helps to classify.

*2.6. Classification Method*

The random forest (RF) algorithm has been applied in many fields of remote sensing and achieved corresponding results [44–47]. It performs random selection of classification features, with random combination of features performed on the nodes of each decision tree. Meanwhile, the multiple decision trees that form the forest can evaluate all features' combination classification results, then select the output classification to evaluate the best results [48]. Compared with support vector machines (SVMs), RF has fewer parameters. It shows the relative importance of different features in classification [49], using random trees to estimate internal error during the training stage. Wang et al. [50] found that the RF method had a higher classification accuracy than the SVM and K-nearest neighbors (KNN) methods in coastal wetland classification. Meanwhile, RF not only improves the classification accuracy of remote sensing images but also is insensitive to noise and over-training [51,52]. Over many attempts, max-depth, max-tree-number, and forest-accuracy were set to 100, 100, and 0.01, respectively.

## 3. Classification Results

Classification of urban vegetation types was carried out in Shanghai and Lianyungang. Vegetation planning and growth in Shanghai are at an advanced level in China, whereas Lianyungang they are at average level. Combining research in Shanghai and Lianyungang can show whether CLBP window textures are stable and universal.

To select optimal CLBP window texture sizes for classification of urban vegetation types, the OA (overall classification accuracy) and KIA (kappa coefficient) of the confusion matrix are used to analyze the overall performance of each CLBP window texture's extraction of urban vegetation. Second, producer accuracy (PA) and user accuracy (UA) are used to analyze the selection of optimal

CLBP window textures for different urban vegetation types. PA refers to the ratio of the number of objects correctly classified as Class A in the whole study area to the actual total number of Class A objects (one column of the confusion matrix). UA refers to the number of objects correctly classified into Class A and the number of objects in the whole research area classified into the total number of class A objects by the classifier (a row of the confusion matrix). Finally, the research results of the two research areas were compared and summarized.

### 3.1. Optimal CLBP Window Texture Size Analysis in Shanghai

Shanghai is one of China's economic centers and has a suitable living climate. Its vegetation planning and growth in Shanghai are among the best in China. Using Shanghai as a research area allows testing of the feasibility of CLBP window textures for classification of urban vegetation types.

### 3.1.1. Classification Performance of Different CLBP Window Textures

By analyzing the OA and KIA of the confusion matrix, overall classification performance for different sizes and types of urban vegetation in Shanghai using a CLBP window texture can be assessed. Table 8 shows the OA and KIA of urban vegetation type classification using different CLBP window textures. In Table 8, the "spectrum" column indicates that only spectral information is used to classify urban vegetation types, whereas the other columns indicate that spectral information and different CLBP window textures are used for classification.

**Table 8.** The OA and KIA of urban vegetation types classification in Shanghai.

|         | $3 \times 3$ | $5 \times 5$ | $7 \times 7$ | $9 \times 9$ | $11 \times 11$ | $13 \times 13$ | Spectrum |
|---------|------|------|------|------|------|------|----------|
| OA (%)  | 66.17 | 63.44 | 61.13 | 65.33 | 57.32 | 48.85 | 48.89 |
| KIA (%) | 55.74 | 54.64 | 52.82 | 57.67 | 48.44 | 38.52 | 38.26 |

As Table 8 shows, all classification results based on a combination of spectral information and CLBP window textures achieved higher accuracy than when using only spectral information. Among the six selected CLBP window texture sizes from $3 \times 3$ to $13 \times 13$, all CLBP texture features except $13 \times 13$ showed markedly improved classification results. The $3 \times 3$ CLBP window texture size achieved the best overall accuracy, reaching 66.17%–17.28% higher than achieved using spectral information alone. The best KIA accuracy obtained by the $9 \times 9$ window size was 57.67% and the classification result improved by 19.41% from spectral information alone.

### 3.1.2. Selection of Optimal CLBP Window Sizes in Shanghai

The selection of optimal CLBP window textures for different urban vegetation types is mainly obtained by PA (Producer Accuracy) and UA (User Accuracy) in the confusion matrix. Table 9 shows the UA and PA of urban vegetation type classification in Shanghai.

Figure 4a–g shows the classification results of urban vegetation types using only spectral information and combining spectral information with $3 \times 3$, $5 \times 5$, $7 \times 7$, $9 \times 9$, $11 \times 11$, and $13 \times 13$ CLBP window textures in Shanghai.

The optimal CLBP window textures in Shanghai can be obtained from PA and UA in Table 9 and classification effect map in Figure 4. The optimal window size of grass and shrubs is $3 \times 3$, and the average value of PA and UA in grass and shrub were 74.10% and 75.60%, respectively. The optimal window size of arbor was $11 \times 11$, and the average value of PA and UA in the arbor was 70.48%. The optimal window size for SG and AG was $9 \times 9$, the average value of PA and UA in SG and AG were 60.84% and 73.00%. The optimal window size for ASG was $7 \times 7$, while the average value of PA and UA in ASG was 75.06%.

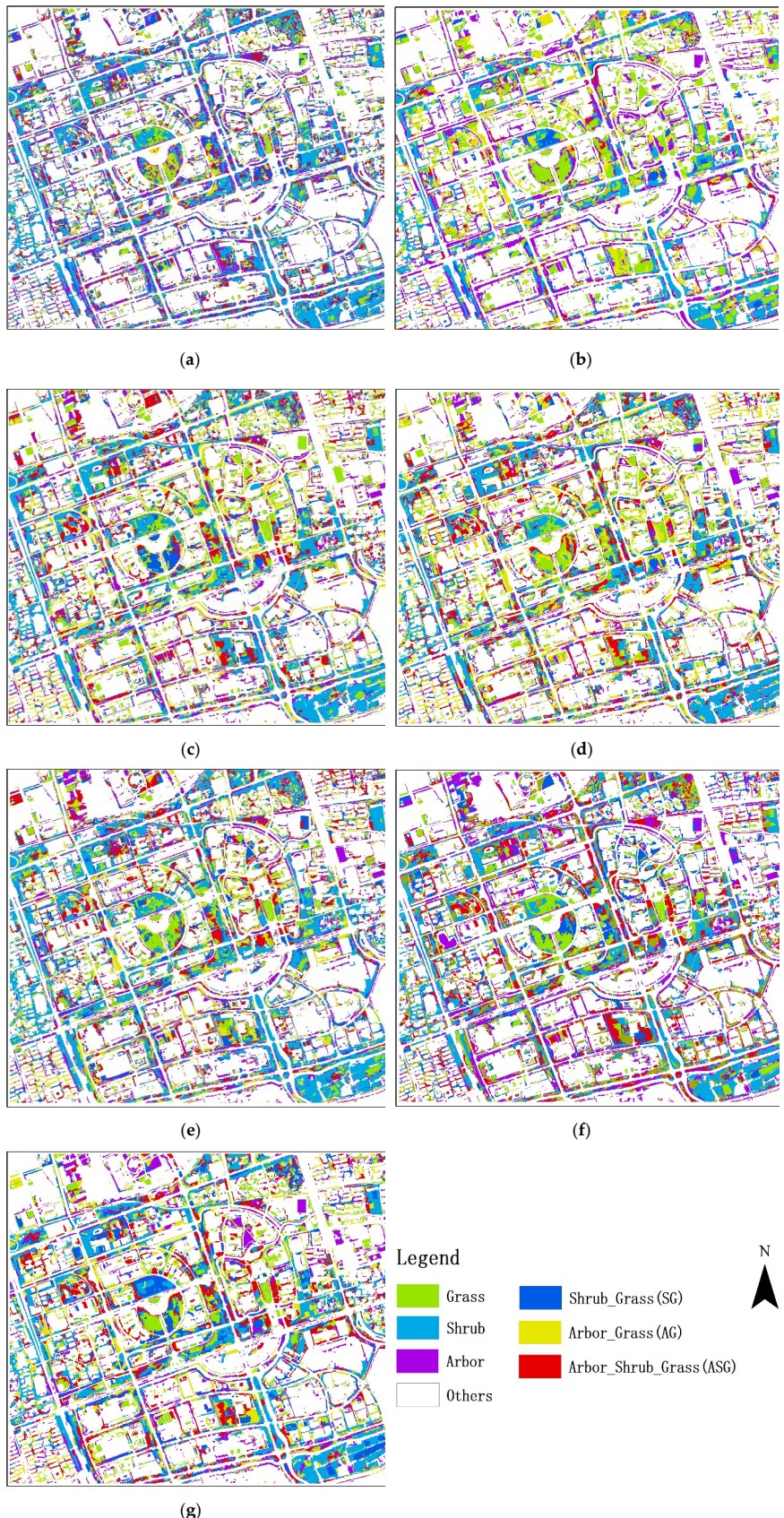

**Figure 4.** Classification map of urban vegetation types in Shanghai based on different CLBP window textures, using only spectral information (**a**) and combining spectral information with CLBP window texture in Shanghai at 3 × 3 (**b**), 5 × 5 (**c**), 7 × 7 (**d**), 9 × 9 (**e**), 11 × 11 (**f**), and 13 × 13 (**g**).

**Table 9.** The PA and UA of urban vegetation types classification in Shanghai.

| PA (%) | Grass | Shrub | Arbor | SG | AG | ASG |
|---|---|---|---|---|---|---|
| 3 × 3 | 78.10 | 75.60 | 72.46 | 29.41 | 54.05 | 8.33 |
| 5 × 5 | 75.76 | 60.87 | 65.52 | 56.25 | 60.60 | 42.86 |
| 7 × 7 | 60.00 | 66.67 | 53.52 | 53.33 | 63.27 | 72.97 |
| 9 × 9 | 54.02 | 81.36 | 48.84 | 66.67 | 78.72 | 67.74 |
| 11 × 11 | 45.00 | 71.43 | 72.46 | 40.48 | 50.00 | 52.50 |
| 13 × 13 | 42.86 | 57.45 | 47.22 | 43.75 | 43.33 | 56.67 |
| **UA (%)** | **Grass** | **Shrub** | **Arbor** | **SG** | **AG** | **ASG** |
| 3 × 3 | 70.09 | 75.60 | 62.50 | 45.45 | 50.00 | 50.00 |
| 5 × 5 | 65.79 | 66.67 | 70.37 | 48.65 | 60.60 | 56.25 |
| 7 × 7 | 65.85 | 77.27 | 62.30 | 50.00 | 44.29 | 77.14 |
| 9 × 9 | 72.30 | 69.57 | 55.26 | 55.00 | 67.27 | 63.64 |
| 11 × 11 | 61.36 | 77.59 | 68.50 | 34.70 | 43.48 | 47.73 |
| 13 × 13 | 47.37 | 67.50 | 53.13 | 37.84 | 52.00 | 37.78 |

*3.2. Optimal CLBP Window Texture Sizes Analysis in Lianyungang*

The Lianyungang area, in the Huaihe River region, has climate characteristics of both North and South China. Its vegetation planning and growth are representative of China overall. Research into classification of urban vegetation types in Lianyungang can verify the universality of CLBP window textures.

3.2.1. Classification Performance of Different CLBP Window Textures

Table 10 shows the overall performance of urban vegetation type classification with different sizes of CLBP window textures in Lianyungang.

**Table 10.** The OA and KIA of urban vegetation types classification in Lianyungang.

|  | 3 × 3 | 5 × 5 | 7 × 7 | 9 × 9 | 11 × 11 | 13 × 13 | Spectrum |
|---|---|---|---|---|---|---|---|
| OA (%) | 59.33 | 54.17 | 61.90 | 64.66 | 57.94 | 52.25 | 57.38 |
| KIA (%) | 50.91 | 44.67 | 54.17 | 57.37 | 48.87 | 42.62 | 48.38 |

As Table 10 shows in the study area of Lianyungang, the 9 × 9 window size achieved the highest overall classification accuracy. The study results also show that the overall accuracy of using only spectrum information is less than using spectrum information and CLBP window textures information. The 9 × 9 CLBP window texture not only used spectral information, but also improved 7.28% in OA and 8.99% in KIA.

3.2.2. Selection of Optimal CLBP Window Sizes in Lianyungang

Table 11 shows the PA and UA of different CLBP window textures under different vegetation types in Lianyungang.

Figure 5a–g shows the classification results of urban vegetation types using only spectral information and combining spectral information with 3 × 3, 5 × 5, 7 × 7, 9 × 9, 11 × 11, and 13 × 13 CLBP window textures in Lianyungang.

Combined with Figure 5 and Table 11, it can be concluded that the optimal window size of grass and shrubs in Lianyungang was 3 × 3, the average values of PA and UA in grass and shrubs were 75.33% and 74.15%, respectively. The optimal window size of the arbor was 11 × 11, while the average value of PA and UA in arbor was 75.10%. The optimal window size for SG and AG was 9 × 9, while the average values of PA and UA in SG and AG were 65.70% and 66.13%, respectively. The optimal window size of ASG was 7 × 7, and the average values of PA and UA in ASG was 73.04%.

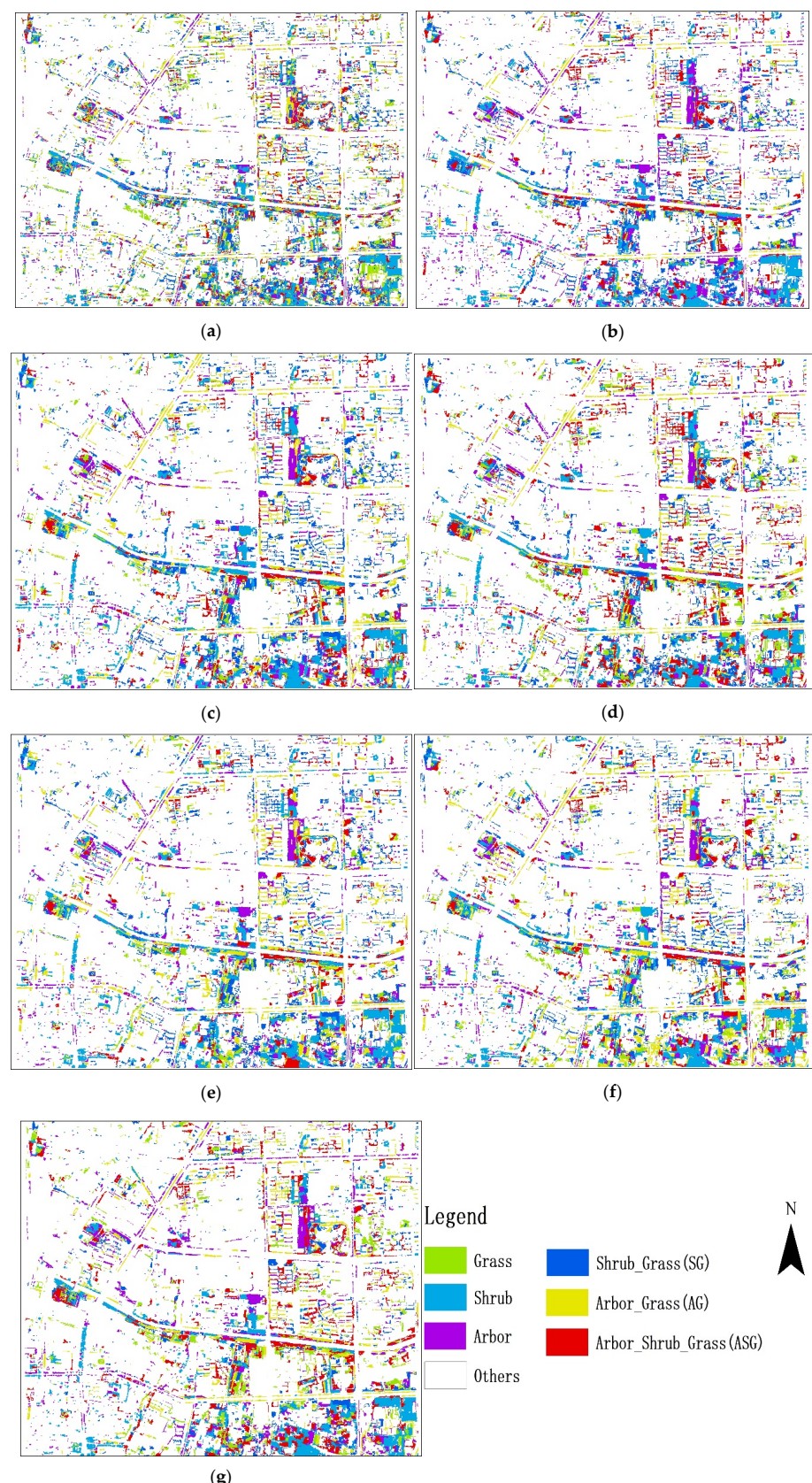

**Figure 5.** Classification map of urban vegetation types in Lianyungang based on different CLBP window textures, using only spectral information (**a**) and combining spectral information with CLBP window texture in Lianyungang at $3 \times 3$ (**b**), $5 \times 5$ (**c**), $7 \times 7$ (**d**), $9 \times 9$ (**e**), $11 \times 11$ (**f**), and $13 \times 13$ (**g**).

**Table 11.** The PA and UA of urban vegetation types classification in Lianyungang.

| PA (%) | Grass | Shrub | Arbor | SG | AG | ASG |
|---|---|---|---|---|---|---|
| 3 × 3 | 68.30 | 68.30 | 58.97 | 56.67 | 45.16 | 51.85 |
| 5 × 5 | 57.14 | 70.00 | 55.56 | 42.42 | 46.88 | 43.33 |
| 7 × 7 | 55.26 | 58.82 | 65.38 | 56.67 | 63.33 | 74.20 |
| 9 × 9 | 62.22 | 53.20 | 70.18 | 70.00 | 73.33 | 56.41 |
| 11 × 11 | 52.08 | 63.33 | 72.41 | 54.17 | 44.44 | 48.15 |
| 13 × 13 | 50.00 | 52.78 | 60.87 | 60.00 | 43.33 | 41.18 |
| **UA (%)** | **Grass** | **Shrub** | **Arbor** | **SG** | **AG** | **ASG** |
| 3 × 3 | 82.35 | 80.00 | 42.60 | 53.13 | 41.18 | 70.00 |
| 5 × 5 | 80.00 | 71.43 | 52.63 | 33.33 | 42.86 | 48.15 |
| 7 × 7 | 53.85 | 68.97 | 65.38 | 54.84 | 59.38 | 71.88 |
| 9 × 9 | 84.85 | 54.35 | 63.50 | 61.40 | 58.93 | 78.57 |
| 11 × 11 | 78.13 | 54.29 | 77.78 | 37.14 | 40.00 | 46.43 |
| 13 × 13 | 67.65 | 59.38 | 59.57 | 32.73 | 56.52 | 48.39 |

### 3.3. Summary of Research Results from Shanghai and Lianyungang

Combining the results of urban vegetation type classification from Figures 4 and 5 and overall classification accuracy from Tables 8 and 10 shows that the vegetation planning and growth effect in Shanghai is better than that in Lianyungang, combined spectral information with CLBP window textures ratio only using spectral information, with classification more accurate than when using only spectral information; overall classification effect of the 9 × 9 window in the two research areas was the best. Tables 9 and 11 indicate that single vegetation has better separability than mixed vegetation, with the overall classification accuracy of single vegetation higher than for mixed vegetation.

Table 12 gives the summary of the experiment and optimal window sizes for the classification of vegetation type in the two study areas. The results show that optimal window size of grass and shrub was 3 × 3; the optimal window size for arbor was 11 × 11; optimal window size of SG and AG was 9 × 9; optimal window size of ASG was 7 × 7. Therefore, the experimental results indicate that the optimal window had a good stability and universality when CLBP window textures were used for urban vegetation types classification.

**Table 12.** The optimal CLBP window sizes.

| Vegetation Type | The Optimal Window Size | Shanghai | | Lianyungang | |
|---|---|---|---|---|---|
| | | PA (%) | UA (%) | PA (%) | UA (%) |
| Grass | 3 × 3 | 78.10 | 70.09 | 68.30 | 82.35 |
| Shrub | 3 × 3 | 75.60 | 75.60 | 68.30 | 80.00 |
| Arbor | 11 × 11 | 72.46 | 68.50 | 72.41 | 77.78 |
| SG | 9 × 9 | 66.67 | 55.00 | 70.00 | 61.40 |
| AG | 9 × 9 | 78.72 | 67.27 | 73.33 | 58.93 |
| ASG | 7 × 7 | 72.97 | 77.14 | 74.20 | 71.88 |

Selection of optimal CLBP window texture for urban vegetation type is related to the characteristics of vegetation texture. Grass and shrub have fine textures, so the optimal window is 3 × 3, whereas the texture of arbor is very rough, for an optimal window of 11 × 11. For mixed vegetation, texture roughness is composed of two or three kinds of roughness ranging from grass to arbor, for an optimal window between 3 × 3 and 11 × 11. It was also found that the optimal CLBP window texture had a limit value: Table 12 shows that arbor has the coarsest texture, with an optimal window size of 11 × 11 instead of 13 × 13.

## 4. Discussion

For urban vegetation type classification, high-resolution remote sensing images can provide more vegetation details. WorldView satellite images and other high-resolution remote sensing images are suitable for classification of urban vegetation types. Compared with use of only spectral information, combination of spectral information with CLBP texture information can improve accuracy of urban vegetation classification by up to 7.28%. In this paper, different sizes of CLBP window texture are discussed to identify the optimal CLBP window for different types of urban vegetation. The results show that selection of optimal window is affected by plant size and vegetation type. The size of the CLBP window texture dramatically affect the accuracy of urban vegetation type classification, so that optimal windows vary among vegetation types.

Compared with other studies, Wang et al. [28] used CLBP window textures to study classification of coastal vegetation types. The optimal window texture was $CLBP_{24,3}$ (circular window), and the accuracy of the $CLBP_{24,3}$ window texture was higher than for GLCM. Optimal windows were selected based only on separability, and the impact of CLBP window texture sizes on accuracy of vegetation type classification was not discussed. Notably, the complexity of the coastal vegetation types studied was much lower than the urban vegetation types. Unlike in the GLCM window texture research, Fu and Lim [39] studied window texture sizes of $3 \times 3$, $5 \times 5$, $7 \times 7$, $9 \times 9$, and $11 \times 11$ for GLCM window textures and found that $7 \times 7$ GLCM window size produced the highest accuracy in litchi tree extraction. Unlike the results of our experiment, the optimal CLBP window size for arbor was $11 \times 11$. Owing to inconsistent texture features and because the litchi tree is not urban arbors, the conclusions reached are inconsistent. Similarly, Yan et al. [19] studied window texture size for GLCM and found that a $3 \times 3$ window size extracted grass in Nanjing with greatest accuracy. The optimal GLCM window size obtained for grass was $3 \times 3$, consistent with the CLBP window texture effect in the experiment, as both studies related to urban grass. The texture characteristics of vegetation types are thus key to determining the optimal texture window size. Woźniak et al. [53] studied SAR image classification using different decomposition windows and image classification accuracies, with $3 \times 3$, $5 \times 5$, $7 \times 7$, and $9 \times 9$ decomposition window sizes used for in-depth research into land use classification accuracy. They reported that for classification of uniform features, the $3 \times 3$ window size can be more accurate: as the window becomes larger, classification details are lost. This conclusion is similar to the results of our experiment, indicating that the texture characteristics determine window size to a degree.

As early as 1992, Woodcock et al. [54] proposed a corresponding window texture size for application to different image feature types. Before 1995, it was concluded that no single window size could be applied to all types of remote sensing objects [55,56], it being inappropriate to use random window sizes for analysis of remote sensing images. The results of this paper validated previous views, while quantitatively clarifying the impact of CLBP window texture sizes on classification accuracy for urban vegetation types. In the body of research into remote sensing images, little research has studied the size of the texture window alone. This paper should prompt scholars to study the impact of window texture sizes on remote sensing image analysis [55,56].

In future research, the shape of CLBP window will be converted from a square window to a circular one. Combining CLBP textures of different window sizes to classify urban vegetation types will allow multiscale window texture sizes to provide more spatial information about vegetation types. Accuracy of CLBP window texture classification of urban vegetation types will be targeted for improvement. In future research, CLBP window textures of different sizes will be used in further remote sensing research.

## 5. Conclusions

Classification of urban vegetation types in Shanghai and Lianyungang produced greater accuracy when using spectrum and CLBP window texture features than when using only spectrum features. For all vegetation types, the $9 \times 9$ window size had the greatest impact on urban vegetation type classification. The $3 \times 3$ window size was optimal for grass and shrubs, $11 \times 11$ for arbor, $9 \times 9$ for

SG (shrub-grass) and AG (arbor-grass), and $7 \times 7$ for ASG (arbor-shrub-grass). It was also found that optimal window size is determined by the roughness of vegetation texture. Classification effects were consistent for the two research areas, showing that the CLBP window texture has stability and universality when used to classify urban vegetation types.

**Author Contributions:** Investigation, Z.C., X.W. and H.Z.; methodology, Z.C., X.W., X.G. and H.Z.; writing—original draft, Z.C. and X.F.; writing—review and editing, J.Y.T., Y.Z., and K.W.; improving the data analysis, K.W., X.F. and Y.Z. All authors have read and agreed to the published version of the manuscript.

**Funding:** This research was funded by the Natural Science Foundation of China (NSFC No. U1901215, 31270745, and 41506106), Lianyungang Land and Resources Project (LYGCHKY201701), Lianyungang Science and Technology Bureau Project (SH1629), the Priority Academic Program Development of Jiangsu Higher Education Institutions (PAPD), the Top-notch Academic Programs Project of Jiangsu Higher Education Institutions (TAPP), and the Marine Special Program of Jiangsu Province in China (JSZRHYKJ202007).

**Acknowledgments:** WorldView-2 satellite imagery data is highly appreciated.

**Conflicts of Interest:** The authors declare no conflict of interest.

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
