# Peer review of "The Influence of CLBP Window Size on Urban Vegetation Type Classification Using High Spatial Resolution Satellite Images"

_remotesensing, doi:10.3390/rs12203393_

Round 1
Reviewer 1 Report
Dear authors, the manuscript is well-conceived, detailed and informative but it has several inadequacies. I shall highlight them and the authors might improve the quality and readability of this research paper accordingly.
- The methodology of this research work should be described by a schematic and conceptual flowchart.
- Summarize the introduction section.
- Please include relevant field photographs of the study area and/or locations.
- All the figures detailing vegetation types and vegetation samples should be geo-referenced.
- All the data-results should be summarized and demonstrated by a graph and a corresponding table.
- Please include and detail all the algorithms and other models mentioned in the manuscript, like CLBP. Please include the relevant references as well.
- The English language of the paper needs a thorough revision. Please have the manuscript proofread by competent authorities and include the certificate on resubmission.
- Please provide all the images in high resolution (For example see Figure 1), the legends are not visible in many cases and hence couldn’t be read properly on several occasions.
- Please make the formatting uniform throughout the manuscript. Currently, it looks quite anomalous.
Author Response
Dear Reviewer:
Attached is the replies to all comments for your checking. We have revised and updated the manuscript based on your comments and suggestions.
Please let us know if any more question.
Yours sincerely,
Yuanzh Zhang
On behalf of all co-authors

Reviewer 2 Report
Dear authors,
in general paper is interesting. However, i find that presentation is rather low. Specific comments are as follows:
Figure 1 is poor quality. Enhance resolution. Also provide coordinates in table format, and mandatory mention coordinate system and ellipsoid. Which color composition is Figure 1? Why is it red?
Line 120 is not correct. Even the flattest surfaces have some curvature. The statement that these conditions are suitable for remote sensing vegetation classification is not correct. Please explain.
In Table 1 reference is missing.
Why did you apply geometric correction if the study area is flat? Which geometric and atmospheric correction is used?
Line 149 what does the large number means?
In Table 2 it is not clear which are spectral features and which ones are texture features. Please adjust the table so it is more understandable.
Formulas 1-3 are missing references.
Table 3 similar problem like Table 2. Try to improve.
In general, training vs inspection samples should be 70-30% in training sample favor (table 5). I believe you would achieve better results if using this recommendation.
Why is it important that RF has less parameters than SVM (line 237).
Line 243 - how many tries? Why use only this parameters? Different parameters provide different results.
Line 254 - UA and PA are not explained.
Line 260 missing point at the end of sentence.
Figure 5 and 6 is not clear, please increase resolution according to Remote Sensing Instruction to authors.
In general it is not clear why are you using different training sizes. Please highlight in text.
Please support conclusions with results (numeric).
Author Response

(The authors gave the same response as above.)

Round 2
Reviewer 1 Report
Dear Authors, I am okay with the changes made. Thanks. You mentioned that 'The manuscript has been proofread by competent authorities and included the certificate on resubmission'. However, I couldn't find the certificate. Please, kindly include it and resubmit the manuscript. Thanks.
Reviewer 2 Report
I believe that authors improved greatly and that now it can be published. Minor technical details will be fixed during publishing period.